# Characterizing Bridge Thermal Response for Bridge Load Rating and Condition Assessment: A Parametric Study

**Artem Marchenko, Rolands Kromanis * and André G. Dorée**

Department of Civil Engineering and Management, Faculty of Engineering Technology, University of Twente, 7522 NB Enschede, The Netherlands; marchenko.artem-@outlook.com (A.M.); a.g.doree@utwente.nl (A.G.D.)
* Correspondence: r.kromanis@utwente.nl

**Abstract:** Temperature is the main driver of bridge response. It is continuously applied and may have complex distributions across the bridge. Daily temperature loads force bridges to undergo deformations that are larger than or equal to peak-to-peak traffic loads. Bridge thermal response must therefore be accounted for when performing load rating and condition assessment. This study assesses the importance of characterizing bridge thermal response and separating it from traffic-induced response. Numerical replicas (i.e., fine element models) of a steel girder bridge are generated to validate the proposed methodology. Firstly, a variety of temperature distribution scenarios, such as those resulting from extreme weather conditions due to climate change, are modelled. Then, nominal traffic load scenarios are simulated, and bridge response is characterized. Finally, damage is modelled as a reduction in material stiffness due to corrosion. Bridge response to applied traffic load is different before and after the introduction of damage; however, it can only be correctly quantified when the bridge thermal response is accurately accounted for. The study emphasizes the importance of accounting for distributed temperature loads and characterizing bridge thermal response, which are important factors to consider both in bridge design and condition assessment.

**Keywords:** structural health monitoring; bridge thermal response; damage detection; bridge static response; measurement interpretation; numerical modelling

## 1. Introduction

Bridges are essential for critical civil infrastructures, allowing traffic to cross difficult terrains and obstacles. Their proper functioning is critical for the local economy and the well-being of society [1,2]. Therefore, effective structural management strategies are necessary. They may not always be achieved with costly and time-consuming visual inspections, which sometimes are performed too late, i.e., when interventions for deteriorated bridge components are needed [1]. Structural health monitoring aims to detect and characterize structural damage in critical civil infrastructure such as bridges, dams and tunnels. Sensors are deployed to collect data, which are then analyzed to assess structural safety, strength, integrity, and performance. Structural damage, for example, can alter stiffness, mass, or energy dissipation properties, affecting the vibrational properties of a structure [3].

Non-physics-based (data-driven) and physics-based approaches co-exist for the structural identification of civil structures [4]. Data-driven approaches use supervised and unsupervised machine learning algorithms to detect damage by extracting damage-sensitive features from the structure's response. Physics-based approaches frequently deploy finite element modelling to identify damage by measuring structural response with a reference model and calibrating, for example, stiffness properties and boundary conditions based on the measured response. Environmental variations such as distributed temperature and solar exposure may cause substantial uncertainties and errors in the computation of the structural response [5].

Dynamic and static responses are affected by changing environmental (e.g., wind, temperature, and humidity [6]) and operational (e.g., mass loadings and operational

speed [5]) conditions. Annual temperature variations may induce strains that are ten times larger than those due to traffic [7]. The identification of bridge response under variations in temperature is important for safety evaluations and condition assessments of bridges [8,9]. Bearings in bridges compensate for both contractions and expansions caused by temperature variations and traffic, allowing for rotation and translation. When the movement is restrained (e.g., locking bearings), structural elements may experience excess stresses, affecting the overall performance, especially movement joints [2,10].

Studies have investigated the impact of temperature variations on dynamic response [11,12]. At sub-zero temperatures, the bridge stiffness and, potentially, its boundary conditions are seen to be altered; for example, increasing values of bridge natural frequencies are observed by the authors of [13]. Ideally, the magnitude of natural frequency is proportional to the stiffness, so when the damage increases, the frequency decreases [14]. In a case when bridge thermal response is not accounted for/characterized, a faulty bridge condition assessment may be given. Moorty and Roeder [15] showed how the Sutton Creek Bridge in Montana, USA, is affected by daily and seasonal temperature variations. The analytical model showed a considerably large expansion of the bridge deck as the temperature increased. For the daily temperature variations, the first mode's natural frequency of the bridge can vary around 5% throughout a 24 h period [16]. For the seasonal temperature variations, the natural frequencies of the bridge can vary around 10% per year for a tested 3-year period [17]. The study on the thermal performance of the Tamar Suspension Bridge in Plymouth, UK, showed that daily and seasonal temperature changes are an important consideration for the serviceability limit state of bridges. Bridge design must account for thermal expansion and contraction cycles. It is important to understand whether the structure has fixed supports and if the structural elements are able to compensate for excessive loads due to expansion or contraction [18].

Mass loading from traffic, such as cars, cyclists, and pedestrians, is an operational variable that induces stress and is difficult to measure. Kim et al. [19] found that measured natural frequencies of a 46 m long steel bridge can decrease by about 5.4% due to heavy traffic. Changes in measured natural frequencies for heavy and light traffic are hardly detectable for mid- and long-span bridges. To estimate the load-bearing capacity and investigate real structural behaviour, finite element (FE) models are normally developed and tested. Using such models and applying static and dynamic load tests, engineers can compare predictions with measured performance benchmarks to obtain insight into making management decisions. However, epistemic uncertainties resulting from temperature changes may be present [10].

Pedestrian footbridges are susceptible to both vertical and horizontal vibrations. This causes resonant responses, which result in high vibration levels and the need for appropriate dynamic design. Generally, vibrations do not cause structural problems; however, they can cause discomfort to its users (i.e., pedestrians) due to exceeding acceleration values. The vertical force is more prominent than the horizontal force; however, both lateral and horizontal components can also cause vibration problems for the structure. The frequency of lateral movement is equal to half of the step of vertical and longitudinal frequency movement [20]. Deviations up to 10% in natural frequencies can occur in a detailed finite element model. The amplitude of natural frequency is influenced by the number of pedestrians. To develop a numerical model and perform dynamic analysis, natural frequencies and vibration modes must be known [21]. In general, walking frequencies for pedestrian bridges are within the range of 1.6 to 2.4 Hz [22,23].

To ensure the structural integrity of the bridge, structural engineers are required to check for deflections and vibrations so that all values meet minimum and maximum value requirements. Deflection is a critical consideration in the serviceability problem and governs the structural design outcome. Deflection limits are used to check whether the maximum allowable deflection limit of the structure is smaller than the measured vertical displacement value [24]. In this context, and in general, vertical displacements are movements of a point on an object relative to the reference point, which are used for the performance and stability assessment of the structure. Studies have focused on

using computer vision-based measurements to obtain influence lines (i.e., bridge response to known loads at specific locations) for the characterization of the load and response mechanisms for condition assessment purposes of bridges [25–27]. Depending on the type of bridge, different maximum deflection limits exist. *Eurocode 3: Design of steel structures* [28] and its supporting part *Design of Steel Structures: Steel bridges* [29] provide no clear guidance on the limit of vertical deflection [30]. Therefore, in this study, a frequently considered limiting standard, $l/500$, where $l$ is the length of the span of vertical deflection, is used [31].

This study investigates temperature effects on the traffic-induced response of a bridge in the context of condition assessment. A range of temperature distribution scenarios and their impact on the response are characterized by static loads and damage. The manuscript is organized as follows. Section 2 lays out the proposed methodology. Section 3 introduces the case study, a footbridge equipped with a SHM system. In Section 4, numerical replicas (in the form of finite element (FE) models) of the bridge are generated. A range of distributed temperature scenarios, static load cases, which could be considered in load ranking tests, and damage are considered. Results, which include damage detection, are presented in Section 5. Section 6 gives the summary, conclusions, and limitations of this study.

## 2. Methodology

To study temperature effects on bridge response, a variety of temperature distribution scenarios must be considered. While performing a condition assessment, it is important to collect both temperature distributions and bridge responses (e.g., vertical displacements and strains) due to static load(s). Then, in each consecutive inspection, the measured response can be compared to the baseline (e.g., no damage) response with, most likely, new temperature distributions. Following a parametric approach, this study develops a methodology for determining changes in bridge response in static tests due to temperatures (and a variety of their distributions) that are not accounted for. Finite element (FE) models are employed, which can give accurate structural responses of under various loading conditions and damage scenarios.

Figure 1 is a flowchart of the methodology introduced in this research. The input to the methodology is temperature distribution ($T$) at the i[th] scenario and the j[th] combination of the nominal traffic load ($L$) (or static load used in a load ranking test). Actions of both (i.e., temperature and static load) need to be understood in order to characterize the bridge response for condition assessment (e.g., recognizing structural damage). Temperature distributions across the structure can directly affect its thermal response [32]. Therefore, bridge response during the baseline and damage conditions must be checked for different temperature distribution scenarios to understand if changes occur. If no changes under the baseline conditions can be observed, there is no need to account for temperature distributions because the bridge response remains unchanged. However, if changes in the response under damage conditions are observed, it is hypothesized that distributed temperature has an effect on it.

The Deformation Area Difference (DAD) Method proposed by Erdenebat et al. [33] can be considered for condition assessment (Figure 2). For long-term bridge condition assessments from inspections, this method can also be used for damage localization. The premise is that baseline structural responses (e.g., vertical deflection) are compared to the newly measured responses. The difference between deflections is presented as deformation area(s). Although the method has been proven to work, the accuracy of its results and their precise interpretation are directly affected by the number of measurement points on the structure [34]. DAD can be determined using Equation (1), which involves squaring the differences in the area between the deflection line curves generated by the analyzed structure state and the corresponding reference system in the absence of damage. In this equation, $f_{d,i}(x)$ represents the deflection function of the damaged curve, and $f_{t,i}(x)$ represents the deflection function of the reference curve. The variable $i$ refers to the specific area section, such as $A_1$, $A_2$, and so on. $\Delta A_i^2$ denotes the area difference in section $i$, while $\sum_1^n \Delta A_i^2$ represents the total area difference in the entire structure [33].

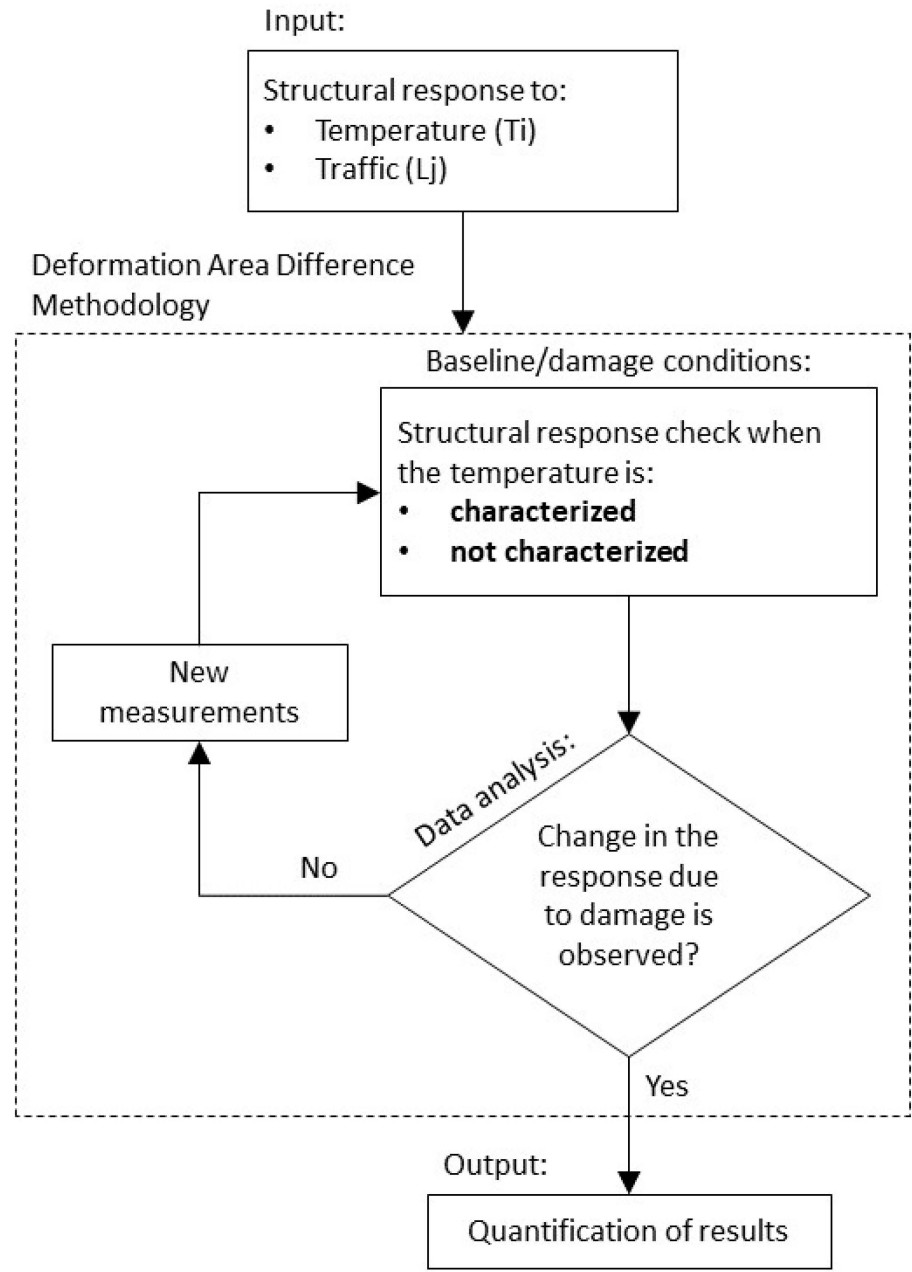

**Figure 1.** Methodology flowchart.

In this study, the traffic load is taken as a UDL applied during the bridge load testing. Under baseline conditions, where only the traffic loads are applied, the response is checked for different temperature distributions. The premise is that vertical deflections due to traffic load would change when a previously unseen temperature distribution is present. Then, changes in the response must be observed and compared as deformation areas when temperature distributions are characterized and not characterized. If no changes are observed for traffic loading only, the new measurements must be collected for other temperature distributions. The same steps must be repeated for new measurements to detect damage. It should be noted that the measured deflections may be very small, although they could fluctuate enough to observe and establish the pattern indicating damage.

$$\mathrm{DAD_i(x)} = \frac{\Delta A_i^2}{\sum_{i=1}^{n} \Delta A_i^2} = \frac{\left[\int_{i-1}^{i} f_{d,i}(x)dx - \int_{i-1}^{i} f_{t,i}(x)dx\right]^2}{\sum_{i=1}^{n} \left[\int_{i-1}^{i} f_{d,i}(x)dx - \int_{i-1}^{i} f_{t,i}(x)dx\right]^2} \tag{1}$$

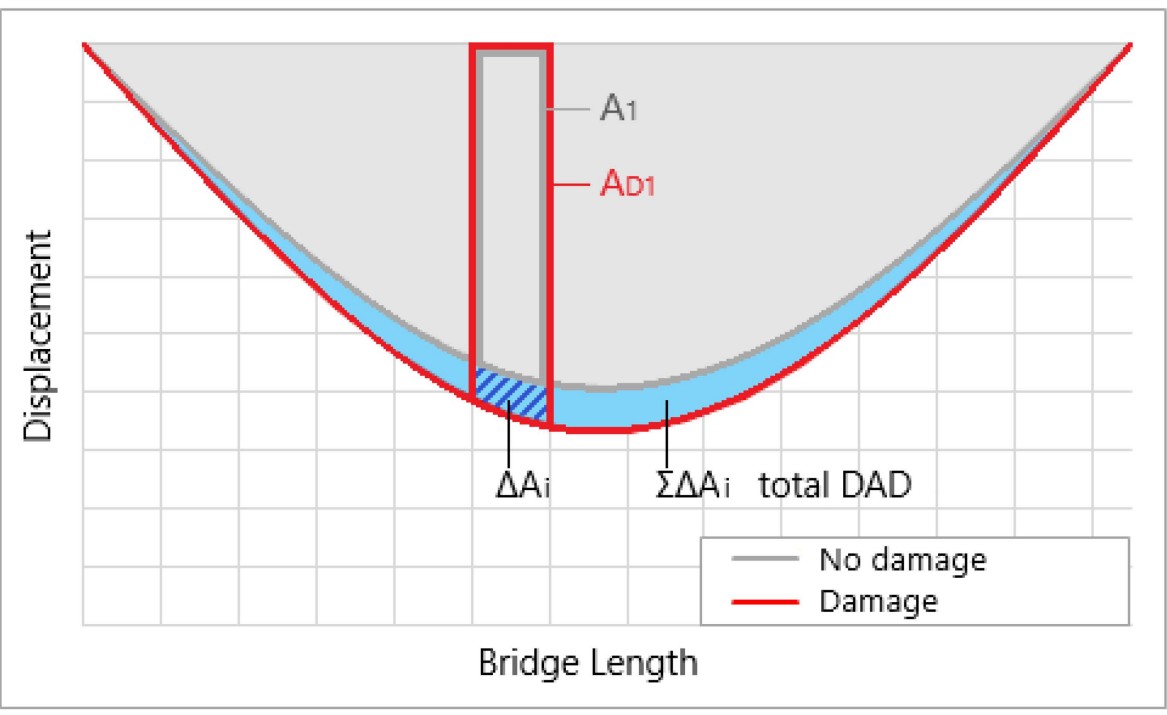

**Figure 2.** The DAD method for condition assessment of bridges.

### 3. The UT Campus Bridge

In this study, the UT Campus bridge (Figure 3) is selected. The bridge serves as a crossing for pedestrians and cyclists over a pond. It is a steel girder bridge located at the University of Twente campus in the Netherlands. The bridge was possibly commenced to the public in the 1980s as a pedestrian footbridge. Spanning 27 m, it connects the Hogekamp (sast) and the sports stadium (west) across a pond. The bridge has some signs of corrosion. The bridge is positioned in the northwest direction. Due to a nearby tall building and trees, only the southwest side of the bridge is directly exposed to the sun, especially during spring and autumn months, resulting in differential thermal expansion/contraction of the structure. The north side of the bridge is mostly hidden from the sun.

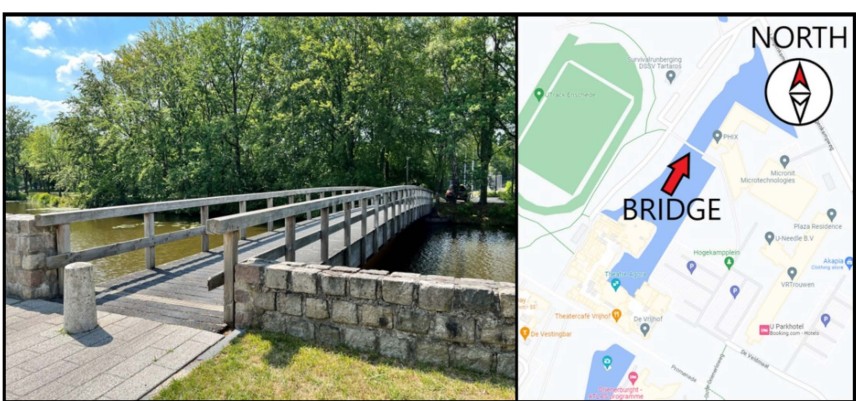

**Figure 3.** The UT Campus bridge from the east side of Hogekamp (**left**) and its geolocation (**right**).

A sketch of the bridge with its main dimensions and selected sensors from its SHM network is shown in Figure 4. The bridge consists of three 27 m long IPE600 steel girders and $80 \times 80 \times 5$SHS horizontal bracings. D40 hardwood oak timber decking and poles with handrails cover the structure. The girders of the bridge are named according to the axes on which they lay. For example, the middle girder, which is on axis B, is named girder B.

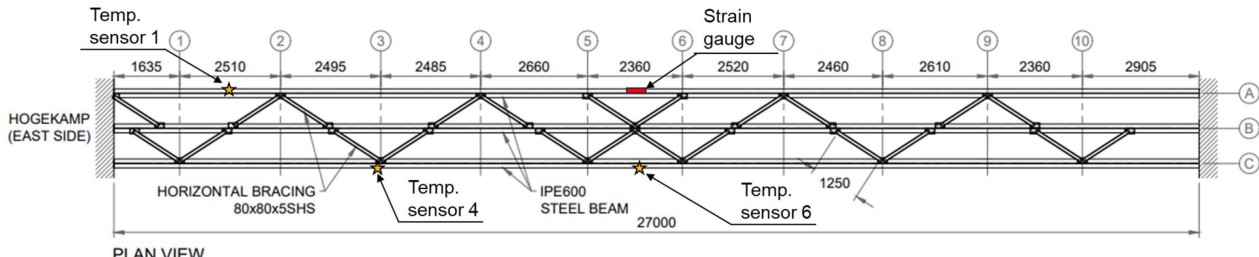

**Figure 4.** A plan view of the UT Campus bridge with selected sensors. Numbers and letters in the circles indicate structural axes.

The sun's path changes with the season. There are two solstices, which are on 2 June and 21 December. During the June solstice, the sun's path gradually moves southwards until December. During the December solstice, the sun's path gradually moves northwards, returning to the celestial equator. The longest day is on the June solstice, and the shortest one is on the December solstice; see Figure 5a,b, respectively. Considering this, the sun's exposure (and the solar radiation that it generates) is different throughout the year for this location.

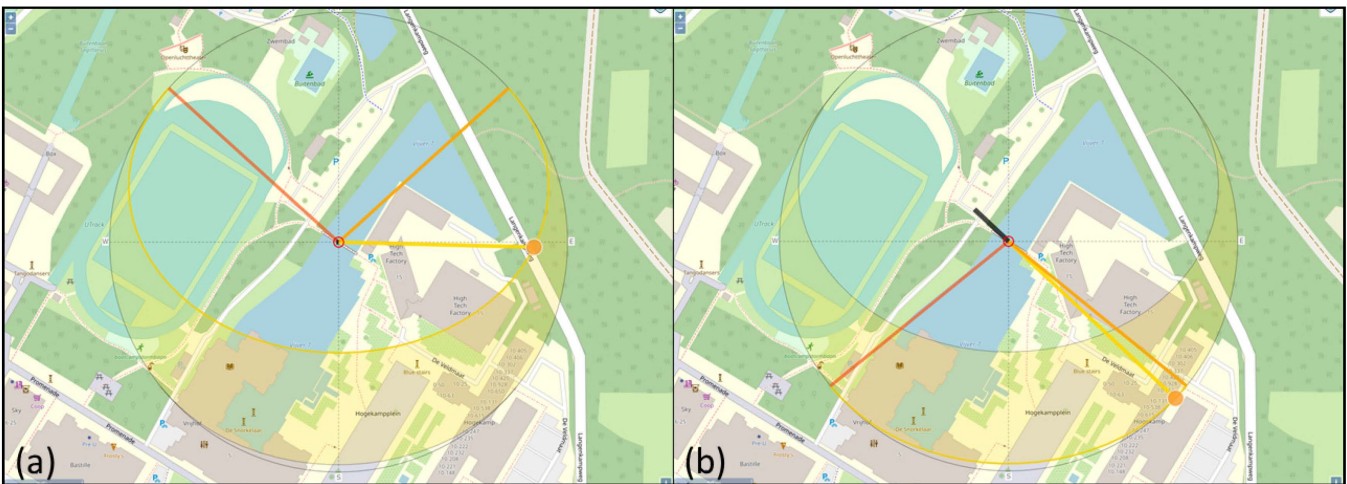

**Figure 5.** Sun path in (**a**) June and (**b**) December.

The bridge is continuously monitored using an SHM system consisting of distributed temperature (Temp.) sensors, accelerometers, strain gauges, and a local weather station. Temp. sensors are surface probes, which have permanent magnets with spring-loaded thermocouples. Figure 6 shows strain and temperature measurements of selected sensors (also shown in Figure 4) for two days. Temperature (Temp.) sensor 1 is installed on the bottom flange of girder A facing south. Temp. sensors 4 and 6 are installed on the bottom flange of girder C, which is always in the shadow, i.e., not exposed directly to the sun. A strain gauge (used in this stud) is installed on the bottom flange of girder A close to the mid-span of the bridge. The relationship between strain and temperature on a sunny day (Figure 6a) suggests that when the temperature increases, the strain tends to decrease. However, when the temperature is constant, such as on a cloudy day (b), the structural response is also constant. Figure 6a shows that material temperature (as measured with temperature sensors) varies across the bridge. Spikes in strain histories present pedestrian and cyclist crossings. Although this is not a high-importance, busy highway bridge, it fits well with the aim of the study, which is the investigation of temperature effects on the traffic-induced response of a bridge in the context of condition assessment. Similar temperature and traffic load-induced responses can be found in the literature [8,35,36].

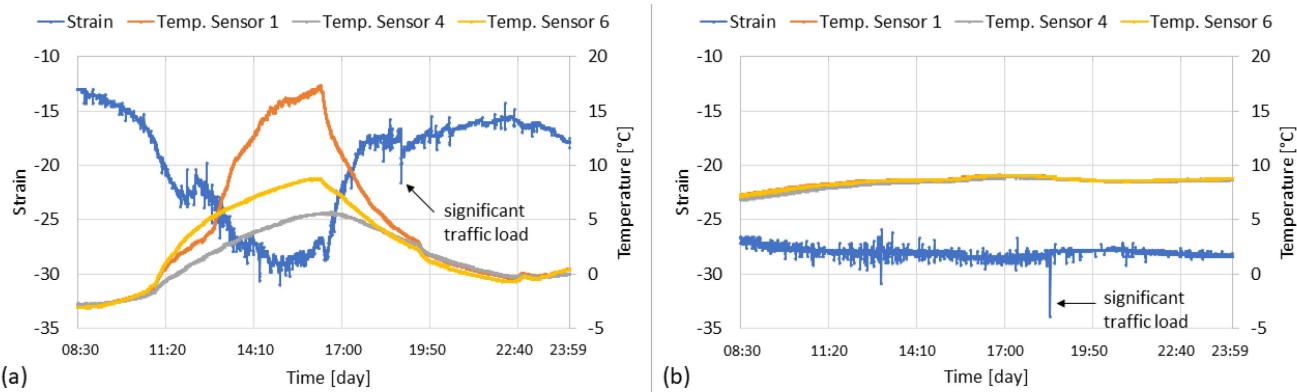

**Figure 6.** Strain and temperature measurements on (**a**) a sunny day and (**b**) a cloudy day.

## 4. Parametric Study

This section presents the parametric study of bridge response. First, the load combinations and temperature distributions are discussed. Last, the process of bridge modelling is expanded on, along with the calibration steps taken to validate the results of performed numerical simulations.

### 4.1. Modelling the Bridge

The numerical replicas of the UT Campus bridge are modelled in Ansys Parametric Design Language (APDL). Ansys is an advanced FE modelling software that supports static and dynamic, structural, and heat transfer analyses [37]. APDL, which is an acronym for "Ansys parametric design language", is a scripting language that was used in conjunction with Ansys to numerically recreate a model [26]. The aim of the numerical models in this study is to generate a bridge response to applied loads that closely matches the expected bridge response. However, all models, in general, are an approximation of reality [10].

A numerical replica of the bridge, as shown in Figure 7, is built setting a unit system that uses metres, kilograms, and seconds. The bridge's geometry (as is) and materials are assigned. The elastic modulus of steel and hardwood is set to 210 GPa and 10.8 GPa, respectively. The reference temperature is set to 20 °C. SHELL181 is selected as an element reference system to model girders and decking. A 1/10 meshing ratio is used. Horizontal bracings are connected to girders using the LINK180 element. In this model, 6309 nodes and 3047 elements are utilized. The geometry of the bridge is slightly simplified. The camber of the deck, which is less than 50 mm, and wooden balustrades are not modelled. Simply supported conditions are applied.

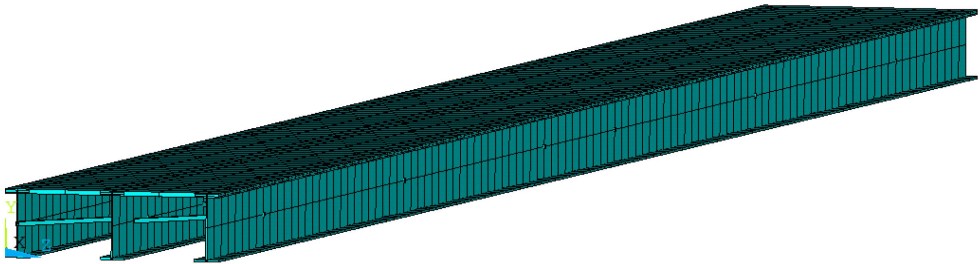

**Figure 7.** FEM of the UT Campus bridge.

### 4.2. Model Calibration

Dynamic model simulations are performed for different boundary conditions to obtain natural frequencies and mode shapes similar to the measured ones [38]. Significant changes in natural frequencies are observed at different combinations of boundary conditions. The models of the two mode shapes and respective natural frequencies are shown in Figure 8. Using the simply supported conditions and restraining only the top flange of the beams at

each side yielded similar modal frequencies to the measured ones. The first two frequencies of the model are 2.47 Hz and 4.23 Hz, corresponding to the first vertical bending and torsional modes. According to Section 5.7, "Dynamic models of pedestrian loads" of Eurocode suggests that forces generated by pedestrians with a frequency matching one of the natural frequencies of the bridge can induce resonance and must be considered in limit state verifications [39]. The frequency 2.47 Hz can be matched by pedestrians crossing the bridge. This, however, has not been considered in the bridge design, which was almost 50 years ago.

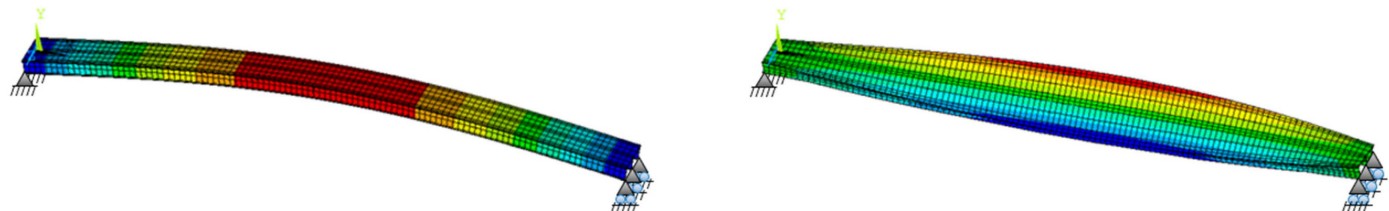

**Figure 8.** Dynamic behaviour of the UT Campus bridge: (**left**) the first vertical bending mode at 2.47 Hz and (**right**) the first torsional mode at 4.23 Hz.

### 4.3. Load Test Scenarios

To characterize operational conditions, two static loading scenarios (Figure 9) are proposed. In the first loading scenario ($L_A$), a uniformly distributed load (UDL) is applied at the centre of the mid-span of the bridge. In the second loading scenario ($L_B$), the same UDL is applied at the side of the mid-span of the bridge. The UDL is set to 4.1 kN/m$^2$. This is based on the characteristic value of UDL in the serviceability limit state (Equation (2)) according to Section 5.3.2.1 "Uniformly distributed load" of EN 1991-2:2003 [39].

$$q_{fk} = 2.0 + \frac{120}{1+30}\left[\text{kN/m}^2\right] \qquad (2)$$

where $q_{fk}$ is the uniformly distributed load, and l is the length of the distributed load. The following check must be made: 2.5 [kN/m$^2$] $\leq q_{fk} \leq$ 5.0 [kN/m$^2$].

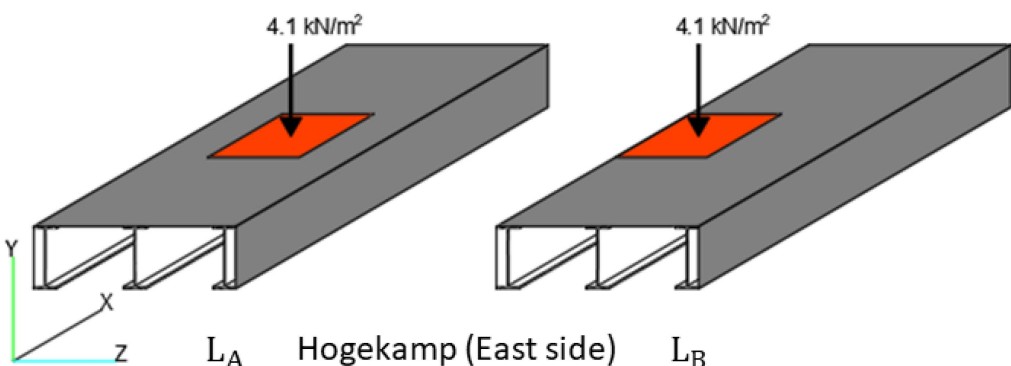

**Figure 9.** Loading scenarios $L_A$ and $L_B$.

### 4.4. Temperature Distributions

Bridges experience complex temperature distributions that are typically non-linear and vary in three spatial dimensions. Depending on whether the local environment has buildings, terrain, or trees, certain parts of a bridge may or may not be fully exposed to solar radiation, hence leading to oblique temperature gradients throughout the day. Not all temperature distributions can be predicted; therefore, specific and extreme temperature distributions are employed to achieve practical design [10]. The availability of accurate and low-cost temperature measurement tools, such as thermal imaging cameras [40] and novel temperature sensors [41], enables the collection of distributed temperatures of bridges.

The UT Campus bridge has a unique position relative to the sun path and the above-mentioned surroundings. Five different temperature distributions are proposed (see Figure 10). These scenarios are derived based on the monitoring data obtained from the SHM system and modified slightly to account for possible extreme future temperature scenarios. The first temperature distribution scenario, T1, represents a cloudy winter day with an ambient temperature of 0 °C. T2 is a hot, cloudy summer day with an ambient temperature of 30 °C. T3 is a linear temperature distribution along the length and depth of the bridge, where temperature drops from 8 to 20 °C. This scenario recreates the sunrise in spring/autumn. T4 is a linear temperature distribution along the width and depth of the bridge; the temperature ranges from 10 to 30 °C. This scenario recreates a hot spring/autumn day at noon. Lastly, T5 is the same as T4 but with a temperature range from 30 to 50 °C. The last scenario represents an extremely hot summer day at noon.

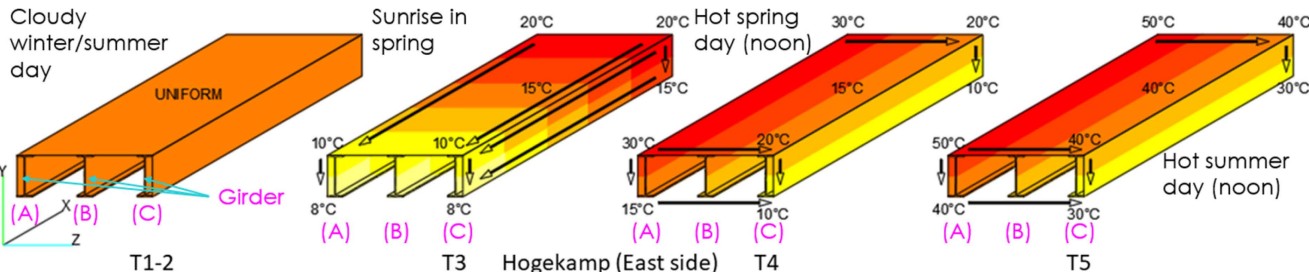

**Figure 10.** Temperature distribution scenarios (T-i, i = 1, 2, 3, 4, 5).

The temperature distribution within the structural elements can be affected by the stored thermal energy. The surface of the structural elements can receive, emit, and reflect shortwave and longwave radiation. Depending on the environmental conditions, the surface can be cooled or heated up by evaporation, precipitation such as rain or snow, melting and freezing processes, influencing heat balance. Heat modelling is based on spatial and temporal interdependent parameters, which are used for the description of environmental conditions and material properties and are important for simulation accuracy [42]. The input material properties for Young's modulus, Poisson's ratio, and thermal expansion coefficients are constant. Hence, the defined material properties are specific to when the ambient temperature is 20 °C.

### 4.5. Bridge Response

Bridge vertical deflections, which can be, for example, collected using computer vision-based systems [43] and image recognition techniques [44], are considered as the bridge response. The response of the girders at load scenarios, $L_A$ and $L_B$, are given in Figure 11. In the first nominal traffic loading combination, $L_A$ resulted in largely equal deflection for all three girders, with Girder B having the largest deflections. However, $L_B$ resulted in different vertical displacements. The deflected shape of girder A increased. Girder B remained unchanged. Lastly, the deflected shape of the girder C decreased.

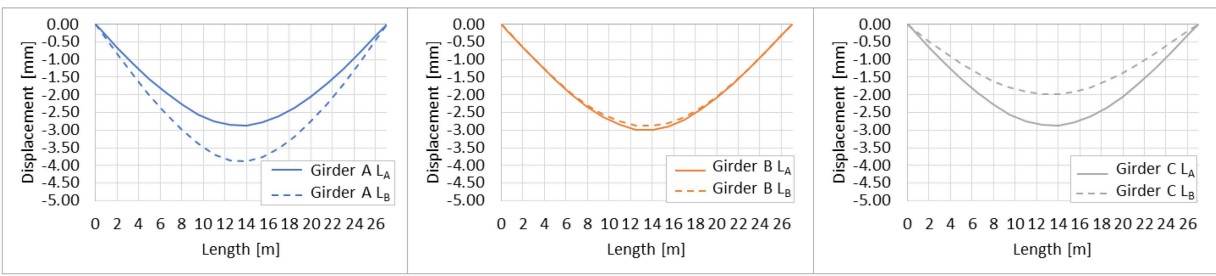

**Figure 11.** Bridge static response for girders A (**left**), B (**middle**), and C (**right**).

An example of temperature load distribution on the FEM and thermal responses due to different temperature distributions are shown in Figure 12. Each temperature distribution is relative to the reference temperature of 20 °C. Although the structure is at rest during the reference temperature, the negative deflection would occur if the load is applied or the average temperature increases. This implies that when a positive deflection occurs, the deflection of the bridge decreases, and when a negative displacement occurs, the deflection increases. In Figure 13, combinations of static and thermal loads are shown. Each traffic loading scenario is applied under different temperature distributions to understand if vertical displacements would deviate significantly. Overall, a similar deflection trend can be observed between two static loads and every temperature distribution. The discussion on vertical displacements is given in the next section.

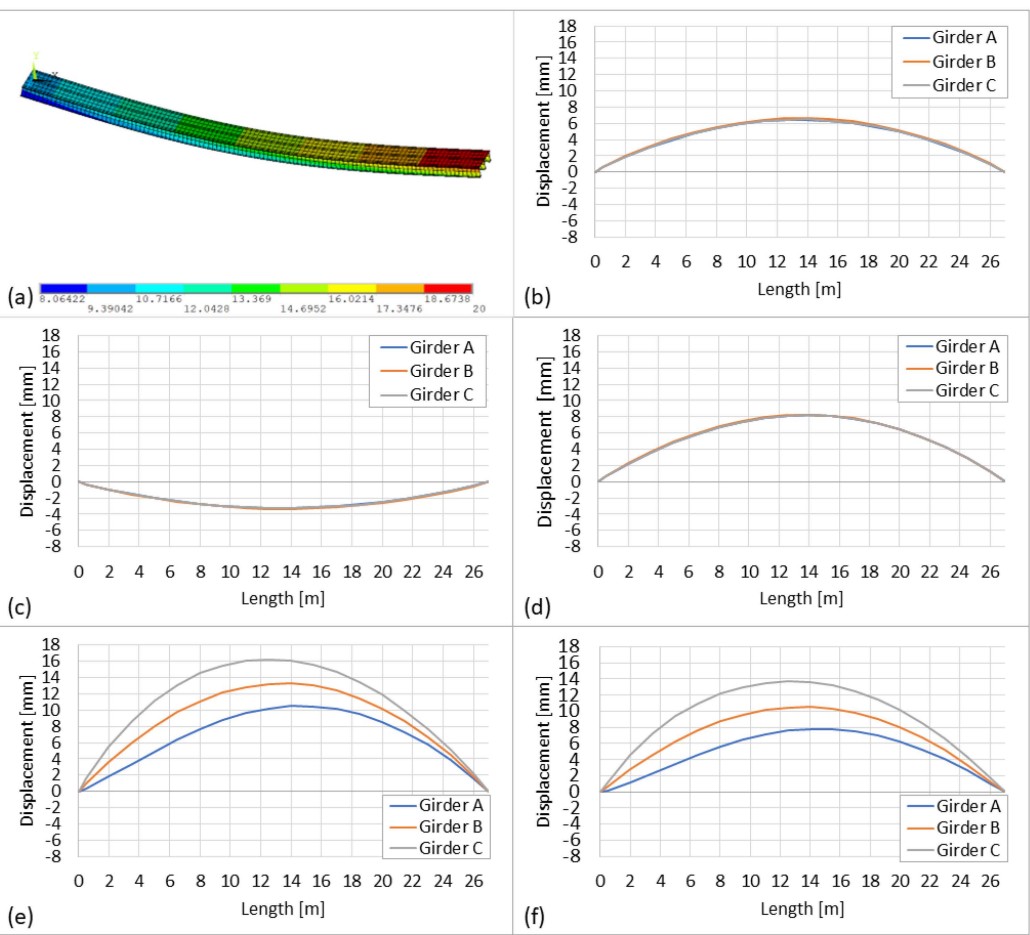

**Figure 12.** (**a**) FEM showing the deformed shape and temperature distribution (scenario T3) of the UT Campus bridge; (**b**–**f**) displacement of girders along the length of the bridge for scenarios T1 to T5, respectively.

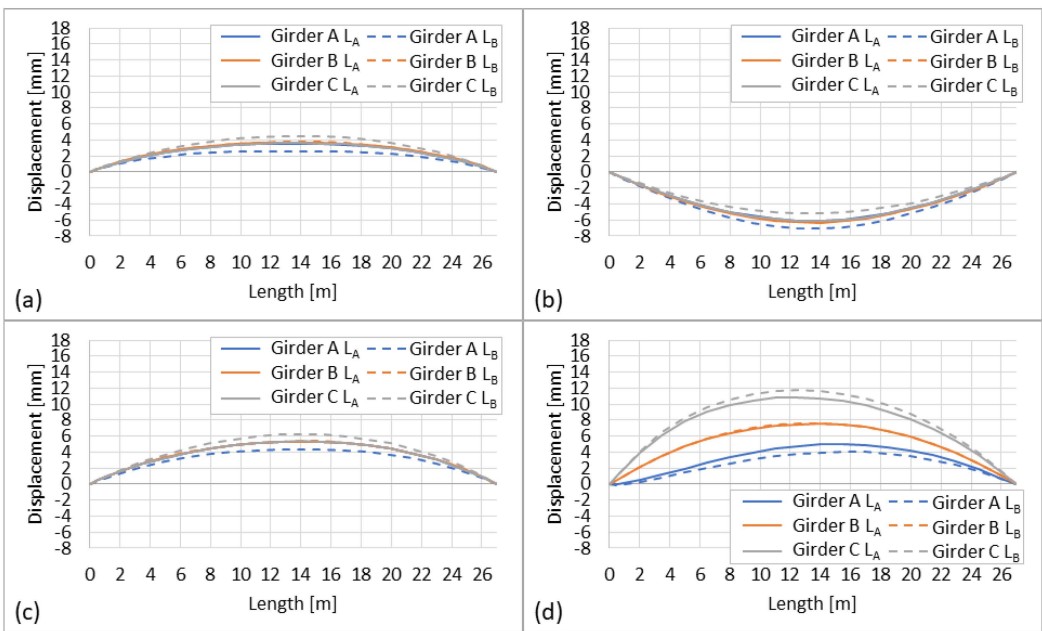

**Figure 13.** Traffic and thermal load responses for (**a**–**c**) for scenarios T1 to T3 and (**d**) for scenario T5, respectively.

### 4.6. Damage

Damage (in the form of steel corrosion) is simulated to study the response of different static and temperature-induced loads. Figure 14 shows the damage severity. Damage is simulated as the reduction in Young's modulus by 10%, 7.5%, and 5% of the bottom flanges of girders A, B, and C, respectively.

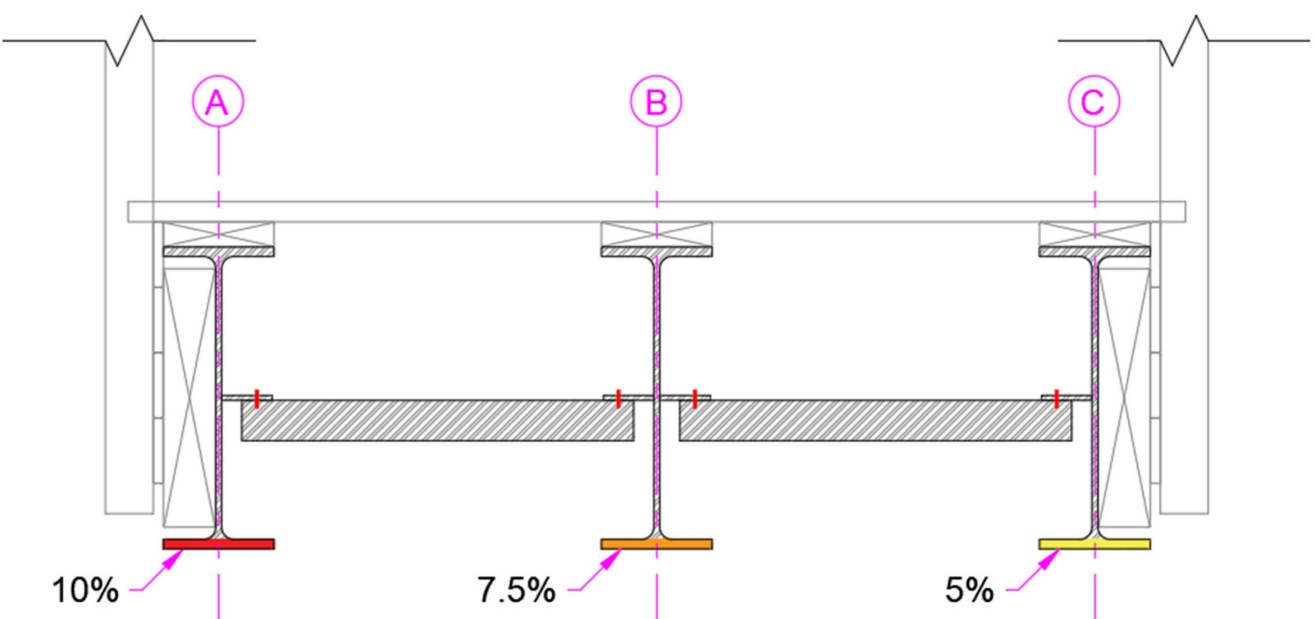

**Figure 14.** Simulated damage shown in the cross-section of the bridge.

## 5. Results and Discussion

### 5.1. Bridge Response

When $L_B$ is applied at the reference temperature of 20 °C, the maximum vertical displacement decreases by 1 mm at girder A and increases by 0.11 mm and 0.89 mm at girders B and C, respectively (see Figure 11). Opposite vertical displacements are at

girders A and C. The change at girder B is insignificant. At T1 (Figure 13a), when the temperature is 0 °C, all girders experience a positive symmetrical deflection of 3.25 mm at the mid-span. At lower temperatures, thermal contraction occurs, leading to an increase in the ultimate tensile strength [45]; therefore, the deflection decreases. At T2 (Figure 13b), when the temperature is 30 °C, all girders experience a negative symmetrical deflection of 6.5 mm at the mid-span. Although temperature changes are rarely uniform, the uniform increase in temperature on the simply supported girder can cause elongation and result in the development of excess stresses [46]. Therefore, the deflection would increase. At T3 (Figure 13c), linear temperature distribution along the length and depth of the structure ranges from 8 to 20 °C. This leads to a large positive symmetrical deflection of 8.2 mm at the mid-span. In T4 and T5, positive unsymmetrical deflections can be observed at the mid-span. T5 is taken as an example (see Figure 13c), where vertical displacements for girders A, B, and C are 7.8, 10.5, and 13.7 mm, respectively. At T4, vertical displacements for girders A, B, and C are 10.5, 13.3, and 16.2 mm, respectively. Girder A experiences the smallest vertical displacements due to extreme temperature distribution, while girder C has the largest due to the lowest temperatures.

*5.2. Damage Detection*

Figure 15 shows deflections of girders before and after the introduction of damage (*D*). The bridge response for each loading combination is slightly different. At the reference temperature of 20 °C, the maximum vertical displacement in girders increases by 3–4%. Due to the nature of the damage (i.e., damage severity reduces from girder A to C), a decreasing pattern in deflections of girders A to C can be observed.

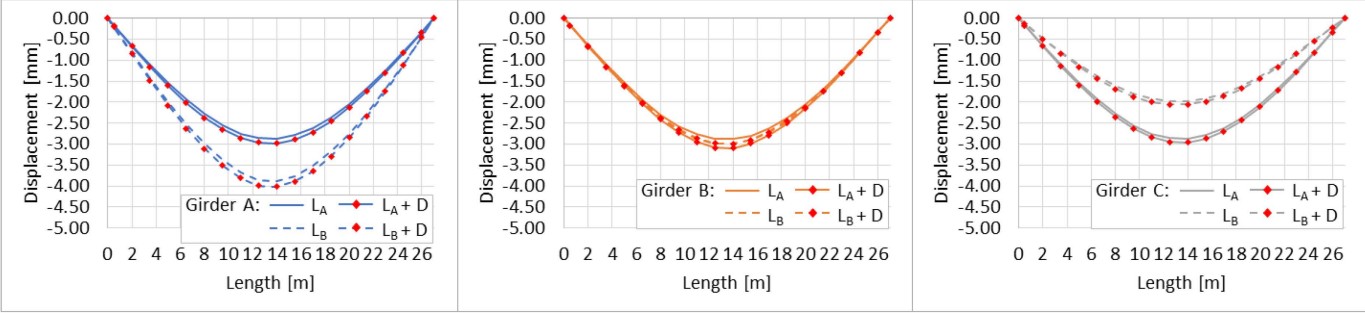

**Figure 15.** Girder A (**left**), B (**middle**), and C (**right**) vertical deflections at baseline and damaged conditions for both loading scenarios.

Figure 16 shows the change in the bridge response under two static load conditions and damage for all temperature distribution scenarios. If temperature distribution yields a negative DAD value, such as in T1, the percentage change is subtracted from the bridge response due to static load and damage at the reference temperature. On the other hand, if a positive DAD value is obtained, such as in T2, the percent change is added to the bridge response due to static load and damage at the reference temperature. It must be noted that the reference values for traffic load $L_B$ deviate significantly from $L_A$ due to the unsymmetrical loading (see Figure 13).

When both traffic and damage are applied, the vertical displacements at T1 and T3 decrease. The change in response at T1 is higher than at T3. At T2, the bridge response increases by a maximum of 5.21% for $L_A$ on girder A and 5.65% for $L_B$ on girder C. Although the temperature distribution is uniform at 30 °C, in $L_A$, girder A experienced a higher increase in deformation of 5.21% than girder B (4.53%),and girder C (4.15%). In $L_B$, girder A has a smaller increase of 4.62% compared to girder B (4.97%) and girder C (5.65%). At T4, the DAD results in both positive and negative responses due to $L_A$ and $L_B$, but no significant changes can be observed. At T5, the DAD results in the highest change in bridge response, yielding a maximum of 5.85% for $L_A$ and 8.10% for $L_B$.

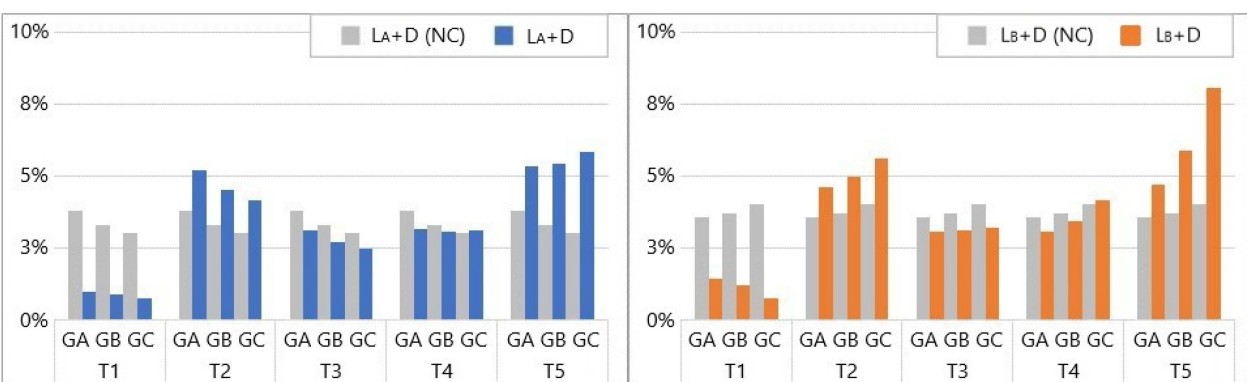

**Figure 16.** DAD for all temperature scenarios and static loads $L_A$ (**left**) and $L_B$ (**right**) at damage state (D). Colour and grey bars give DAD results for conditions with the thermal response characterized and not characterized (NC).

For every consecutive inspection in Figure 16, the deformation areas were found to be different for traffic load and damage when the effects of thermal response were removed. Results show that deformation areas of traffic load and damage increased for all girders by an average of 0.99–0.77% in T1, 5.21–4.15% in T2, 3.13–2.47% in T3, 3.18–3.12% in T4, and lastly, 5.35–5.85% in T5. The deformation areas of traffic load and damage increased for girders by 1.46–0.77% in T1, 4.62–5.65% in T2, 3.06–3.21% in T3, 3.10–4.15% in T4, and lastly, 4.73–8.10% in T5.

*5.3. Discussion*

The bridge response to damage (when the static load is applied, and damage is created) decreases when the temperature is uniform at 0 °C. If the temperature is higher than the reference temperature by 10 °C at T2 (30 °C), the response increases. The highest change in the bridge static response is observed when static load and damage are simulated during extremely hot summer days (30–50 °C). Varying temperature distribution, 10–30 °C, results in varying deformation areas, yielding both an increase and decrease in the response. The smallest changes in DAD are observed for T3 and T4. These are the scenarios when the overall temperature difference between the reference temperature and the temperature distribution in the set scenario is small, i.e., around 10 °C.

$L_A$ shows similar responses throughout all girders. When the thermal response is not characterized and the damage is present, in the parametric simulation, the bridge response values change. Therefore, inaccurate conclusions about the structure's state can be made. The results show that unsymmetrical loads (load scenario $L_B$), which cause torsion in the bridge, influence the temperature effect by making it increase the magnitude of the response of the unloaded side at extreme temperatures by 8%. The higher the temperature, the larger the response of the unloaded side.

## 6. Conclusions

Responses of bridges are unique and governed by temperature distributions, which may cause major deformations that are larger than or equal to peak-to-peak traffic loads in the long term. Therefore, correct characterization of the bridge thermal response is important in condition assessment. This study uses numerical replicas (in the form of finite element (FE) models) of the UT Campus bridge to show that the relationship between distributed temperatures in the bridge response under static loads (such as during load testing) and damage is different. The following conclusions can be made:

- The bridge response to the static load and damage can either increase or decrease depending on temperature distribution when its response is neglected, leading to inaccurate conclusions about the structure's conditions.

- Simplistic assumptions such as neglecting temperature-induced response can lead to errors of up to 8% under extreme temperature distributions.
- When an unsymmetrical load is present and results in torsion (of a bridge), the thermal response can increase as much as twice at the unloaded side.
- Changes in deformation areas can be small (~0.77%), but when correctly accounted for, their cumulative summation may reveal areas of damage.

During the design and inspection of bridges, it is important to account for temperature effects. When the bridge is damaged, its response changes. Bridge responses such as vertical displacements can be larger than the calculated values when assuming normal (e.g., no damage) conditions. Therefore, knowledge of the temperature effect on the bridge response when employing the parametric models can support the condition assessment of bridges. Although the study demonstrates that (i) the bridge response varies along with distributed temperatures and (ii) it is important to characterize bridge thermal response for accurately assessing bridge conditions, results solely from numerical models of a bridge are deployed. There is a need to validate the premise of the research on laboratory test beds or, ideally, full-scale bridges. Further work is required to fully understand and investigate the complexity behind changes in the bridge thermal response. In addition, FE models can be enhanced by comprehensive climatic data to account for a variety of cooling or heating scenarios of the bridge surface via evaporation, precipitation, melting and freezing processes that affect the overall heat balance [42].

**Author Contributions:** Conceptualization, A.M., R.K. and A.G.D.; methodology, A.M. and R.K.; modelling and data analysis, A.M.; writing—original draft preparation, A.M.; writing—review and editing, A.M. and R.K.; visualization, A.M.; supervision, R.K. and A.G.D. All authors have read and agreed to the published version of the manuscript.

**Funding:** This research received no external funding.

**Institutional Review Board Statement:** Not applicable.

**Informed Consent Statement:** Not applicable.

**Data Availability Statement:** The data that support the findings of this study are available upon reasonable request.

**Conflicts of Interest:** The authors declare no conflicts of interest.

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
