# Peer review of "Characterizing Bridge Thermal Response for Bridge Load Rating and Condition Assessment: A Parametric Study"

_infrastructures, doi:10.3390/infrastructures9020020_

Round 1

Reviewer 1 Report

Comments and Suggestions for Authors

This investigation mainly studies the thermal response characteristics of bridges and its influence on the load assessment and condition assessment of bridges. The manuscript is well written and presented; however please discuss or incorporated the following comments.

In this paper, a pedestrian bridge with pedestrian and bicycle loads is selected, and some conclusions on thermal response and damage detection of the bridge are presented, and how to ensure the universality and reliability of these conclusions for other Bridges?

It can be seen in Figure 3 that the UT Campus bridge has a certain camber. How is it considered in the FEM?

Line 195 and Figure 6, How is the SHM system laid out and should an explanation be given?

Line 224: Is it reliable to use only frequency for model calibration, and is it considered to supplement the verification by loading experiment?

Line 231 and 179:Using the simply supported conditions and restraining only the top flange of the beams”,“The boundary conditions are assumed to be fixed.” Whether the two sentences are contradictory?

Reviewer 2 Report

Comments and Suggestions for Authors

This research evaluates the significance of characterizing the thermal response of bridges and distinguishing it from the response induced by traffic. The proposed methodology is validated using numerical replicas, specifically fine element models, of a steel girder bridge. While the paper is intriguing, it needs some adjustments before it can be published.

·       In the line 53, the authors discuss the importance of monitoring temperature in studying the dynamic responses of bridges. The authors are encouraged to present more advancements in consideration this parameter in Structural Health Monitoring (SHM). In fact, recently novel low-cost monitoring systems have been developed to determine the dynamic response of structures (https://doi.org/10.3390/su13073695) as well as deriving temperature parameters in structures (https://doi.org/10.1016/j.heliyon.2023.e17282).

·       The gap in the literature and the principal objective/motivation of the paper are not well presented.

·       In line 217 were the authors are discussing the bridge model, it is interesting to mention the number elements used for the bridge discretization.

·       Did the authors perform any mesh sensitivity analysis? Providing this discussion would be interesting.

·       Discussion over the results of the paper is so limited.

·       Limitation of the study may be presented at the end of the paper.

·       Future possible research may be indicated in the end of the Conclusions.

Reviewer 3 Report

Comments and Suggestions for Authors

This article uses the finite element method to study the effect of different temperature distributions on the response of a steel girder bridge under damage and static loads. The author first proposed a method for determining changes in bridge response based on the Deformation Area Difference method. Next, use Ansys Parametric Design Language for finite element modeling according to the design drawing of the steel girder bridge. The author proposed two different static loads and five different temperature distributions and conducted experimental research on the simulated damage of three girders of the bridge. Research has shown that the relationship between distributed temperature loads in the bridge response, under static loads and damage is different.

1. Figure 6 mentions three sensors. Please provide a detailed explanation of where these three temperature sensors are installed on the bridge.

2. From lines 199 to 120 of the text, it is mentioned that according to Figure 5, it can be seen that the material temperature of the bridge is different. Can the author explain how this is observed?

3. The use of different boundary conditions for simulation was mentioned in lines 227 to 228 of the text, but only the simple support condition was simulated, which contradicts the previous description.

4. Section 4.4 of the article mentions the selection of temperature distribution and material temperature but does not provide a basis for selection. Please provide a detailed introduction to What is the basis for selecting the five temperature distributions and material temperatures.

5. One relative study should consider to be mentioned: Fiber optic health monitoring and temperature behavior of bridge in cold region, Structural Control and Health Monitoring.

6. According to Figure 16, under T4 conditions, the DAD values of the three main beams during thermal response characterization are either lower or higher than those without characterization. Could the author please explain why this phenomenon occurs?

7. The data from Table 1 and Table 2 are mentioned in lines 366 to 372 of the text, but Table 1 and Table 2 are not provided in the text. Please add these two tables.

Round 2

Reviewer 2 Report

Comments and Suggestions for Authors

The authors have applied all the proposed comments which leaded to enhance the quality of the paper, sufficiently. Thus the paper can be proceeded with the publication process.

Reviewer 3 Report

Comments and Suggestions for Authors

The manuscript has been revised and the quality has been improved.